# Diagnostic Performance of Bronchoalveolar Lavage (1,3)-β-d-Glucan Assay for *Pneumocystis jirovecii* Pneumonia

**DOI:** 10.3390/jof6040200

**Published:** 2020-10-01

**Authors:** Shiwei Zhou, Kathleen A. Linder, Carol A. Kauffman, Blair J. Richards, Steve Kleiboeker, Marisa H. Miceli

**Affiliations:** 1Division of Infectious Diseases, Department of Internal Medicine, University of Michigan, Ann Arbor, MI 48109, USA; shzh@med.umich.edu (S.Z.); linderk@med.umich.edu (K.A.L.); ckauff@med.umich.edu (C.A.K.); 2Infectious Diseases Section, VA Ann Arbor Healthcare System, Ann Arbor, MI 48105, USA; 3Michigan Institute for Clinical & Health Research, University of Michigan, Ann Arbor, MI 48109, USA; blairr@med.umich.edu; 4Viracor Eurofins Clinical Diagnostics, Lee’s Summit, MO 64082, USA; SteveKleiboeker@viracor-eurofins.com

**Keywords:** *Pneumocystis* pneumonia, β-d-glucan, bronchoalveolar lavage fluid

## Abstract

We evaluated the performance of the (1,3)-β-d-glucan (BDG) assay on bronchoalveolar lavage fluid (BALF) as a possible aid to the diagnosis of *Pneumocystis jirovecii* pneumonia. BALF samples from 18 patients with well-characterized proven, probable, and possible *Pneumocystis* pneumonia and 18 well-matched controls were tested. We found that the best test performance was observed with a cut-off value of 128 pg/mL; receiver operating characteristic/area under the curve (ROC/AUC) was 0.70 (95% CI 0.52–0.87). Sensitivity and specificity were 78% and 56%, respectively; positive predictive value was 64%, and negative predictive value was 71%. The low specificity that we noted limits the utility of BALF BDG as a diagnostic tool for *Pneumocystis* pneumonia.

## 1. Introduction

The definitive diagnosis of *Pneumocystis jirovecii* pneumonia requires visualization of the organism using silver or fluorescent antibody staining on bronchoalveolar lavage fluid (BALF) or lung tissue. The EORTC/MSGERC consensus definitions for invasive fungal infections note that a positive BALF polymerase chain reaction (PCR) or a positive serum β-d-glucan (BDG) allows a diagnosis of probable *Pneumocystis* pneumonia [1]. Several studies and meta-analyses have noted a sensitivity of greater than 95% when serum BDG is used for the diagnosis of *Pneumocystis* pneumonia in patients with HIV [2,3,4]. When other immunocompromised hosts have been studied, the serum BDG assay has been reported to have a sensitivity varying from 86% to 94%, perhaps reflecting a lesser burden of organisms in the non-HIV population [4,5,6,7,8,9]. In all of these studies, results varied when different cut-off points for positivity were used.

We were interested in examining the utility of quantitating BDG in BALF and were especially interested in assessing the value of quantitating BALF BDG values ≥500 pg/mL. We undertook a single-center, retrospective case-control study to determine the performance characteristics of BDG in BALF with a view to the possible utility of using this assay as an aid in the diagnosis of *Pneumocystis* pneumonia.

## 2. Patients and Methods

### 2.1. Patients and Setting

This study was performed at the University of Michigan Health System and was approved by the University of Michigan and the VA Ann Arbor Healthcare System Institutional Review Boards. Residual BALF obtained at the time of bronchoscopy on adult patients were prospectively collected from November 2015 to February 2018 and stored in aliquots at −70 °C in the VA Infectious Diseases Laboratory.

Patients who had a positive BALF PCR for *P. jirovecii* and who were diagnosed with proven, probable, or possible *Pneumocystis* pneumonia were included in the study; they were matched 1:1 by age within 5 years and by risk factors with control patients who did not have *Pneumocystis* pneumonia. BDG results were not included in this categorization. Patients with a positive BALF PCR who did not meet criteria for proven, probable, or possible *Pneumocystis* pneumonia were excluded from further analysis.

Factors, such as concomitant Gram-negative bacterial pneumonia and colonization of the tracheobronchial tree by fungi that might impact the performance of the BALF BDG, were recorded. Patient medical records were reviewed, and data were entered into a REDCap^®^ database.

### 2.2. Definitions

The categorization of proven, probable, or possible *Pneumocystis* pneumonia was based on strict diagnostic criteria related to host factors, clinical presentation, radiologic studies, histopathological findings, and response to therapy [10]. Host factors included: HIV (CD4 < 200/μL or <14%); high dose corticosteroids equivalent to prednisone dose >20 mg daily for >30 days; untreated Cushing’s Disease; solid tumor with chemotherapy within 3 months; hematologic malignancy; solid organ transplant; primary or acquired T-cell/B-cell immunodeficiency; and treatment with rituximab, alemtuzumab, or TNF-α inhibitor within 6 months, purine analogs, alkylating agents, or rapamycin inhibitor within 90 days, or small molecule kinase inhibitors within 1 year. Clinical presentation compatible with *Pneumocystis* pneumonia included new onset or worsening of cough, dyspnea, and hypoxia (defined as new or increased need for >2 L supplemental oxygen from baseline), with or without fever. Radiological studies that were required were new radiographic or computerized tomographic (CT) findings of diffuse infiltrates, ground glass opacities, or nodules. Histopathological criteria for the diagnosis of *Pneumocystis* pneumonia were visualization of the organism using silver or fluorescent antibody staining on BALF or lung tissue.

Based on the above diagnostic criteria, patients were categorized as having proven, probable, or possible *Pneumocystis* pneumonia. Patients with proven *Pneumocystis* pneumonia had to meet clinical presentation and radiographic criteria and had identification of *Pneumocystis* organisms by silver stain or specific immunofluorescence stain on BALF or lung tissue; host factors could be present or not. Probable *Pneumocystis* pneumonia was defined as those patients who had at least one host factor, met clinical and radiographic criteria, showed response to treatment for *Pneumocystis* pneumonia, and no alternative diagnosis was found. Possible *Pneumocystis* pneumonia included patients who met clinical and radiographic criteria, host factors could be present or not, response to treatment for *Pneumocystis* pneumonia may or may not have happened, and no alternative diagnosis was found. Patients who had a positive *Pneumocystis* PCR test but who did not meet the diagnostic criteria noted above and who had an alternative diagnosis were classified as not having *Pneumocystis* pneumonia.

### 2.3. P. Jirovecii PCR Testing

#### 2.3.1. Endpoint PCR

BALF specimens collected prior to 1 February 2017, were tested by an in-house *Pneumocystis* endpoint PCR assay. The target amplified in this procedure was the mitochondrial ribosomal RNA gene *rnl* (large subunit ribosomal RNA). The amplification product was visualized after being run on an agarose gel and then stained with ethidium bromide. To be called positive, a band had to be present at 346 base pairs. Positive and negative controls were always concomitantly tested with patient samples to ensure the test’s validity.

#### 2.3.2. Real-Time PCR

Beginning 1 February 2017, the EliTech InGenius PCR assay (EliTech Group, Bothell, WA, USA) was used for the detection of *P. jirovecii* DNA in BALF samples. The target amplified in this procedure was the mitochondrial ribosomal RNA gene *MtLSU*. Specimen extraction, purification of nucleic acids, amplification, and detection of *Pneumocystis* DNA were all performed by the InGenius instrument. Readouts provided by the instrument showed amplification curves and cycle threshold values. Positive and negative controls were always concomitantly tested with patient samples to ensure the test’s validity.

### 2.4. BDG Testing

BDG testing of BALF was performed on de-identified coded samples at Viracor Eurofins Laboratory using the Fungitell^®^ assay (Beacon Diagnostics, Associates of Cape Cod, Falmouth, MA, USA). BALF BDG assays were performed in the same manner as serum BDG assays according to the manufacturer’s instructions. In brief, glucan standards were serially diluted to 500, 250, 125, 62.5 and 31.25 pg/mL. An alkaline pre-treatment solution (0.125 M KOH/0.6 M KC1) was added to each sample. If the reading was >500 pg/mL further serial 1:10 dilutions were performed in reagent grade water. Agitation was performed for 5–10 s, and the plate was then incubated for 10 min at 37 °C. The Fungitell^®^ reagent was reconstituted and added to samples, controls, and standards. The plates were placed in a microplate reader (equilibrated to 37 °C) and read at 405 nm minus 490 nm, for 40 min at 37 °C. The mean rate of optical density change (milli-absorbance units per minute) for all points between 0 and 40 min was interpolated with results from the standard curve to determine the quantity of (1,3)-β-d-glucan. Negative and positive controls were included in each assay run. The BALF BDG internal validation cutoff was ≥45 pg/mL.

### 2.5. Statistical Analysis

Sensitivity, specificity, positive predictive value (PPV), and negative predictive value (NPV) for the diagnosis of *Pneumocystis* pneumonia were determined for BALF BDG. Receiver operating characteristic/area under the curve (ROC/AUC) analyses were used to determine the optimal cutoff value of BALF BDG that might be predictive for the diagnosis of *Pneumocystis* infection. Statistical analysis was performed using SAS 9.4 statistical software.

## 3. Results

Eighteen patients had a positive *P. jirovecii* PCR test and were categorized as having proven (*n* = 4), probable (*n* = 8), and possible (*n* = 6) *Pneumocystis* pneumonia. These 11 men and seven women with a mean age of 59.2 ± 13.7 years had host factors that included high dose corticosteroids (*n* = 6), solid organ transplant (*n* = 4), T-cell immunosuppressive medications (*n* = 4), active malignancy (*n* = 2), and HIV infection (*n* = 1). The matched control cohort had 12 men and six women, with mean age 59.5 ± 13.9 years. Host factors included high-dose corticosteroids (*n* = 6), solid organ transplant (*n* = 4), T-cell immunosuppressive medications (*n* = 4), active malignancy (*n* = 2), and HIV (*n* = 1). Two patients in each cohort had no typical risk factors for *Pneumocystis* pneumonia but were critically ill and underwent testing for *Pneumocystis* pneumonia.

The median and range of BDG values were 465 (<45–10,400) pg/mL in the *Pneumocystis* cohort and 95 (<45–5840) pg/mL in the control cohort (Figure 1). Higher BDG values were not predictive of *Pneumocystis* infection and did not differentiate proven *Pneumocystis* pneumonia from possible *Pneumocystis* pneumonia. Two patients with probable *Pneumocystis* pneumonia, one with proven *Pneumocystis* pneumonia, and one with possible *Pneumocystis* pneumonia all had BDG values >3000 pg/mL. The best performance was observed when the cut-off value for BALF BDG was 128 pg/mL; ROC/AUC analysis showed AUC 0.70 (95% CI 0.52–0.87) (Figure 2). Sensitivity was 78%, specificity was 56%; the PPV was 64%, and the NPV was 71%. Eight control patients had BDG levels higher than this cut-off, and six had BALF BDG values ≥500 pg/mL. When only the 12 proven/probable *Pneumocystis* pneumonia cases and their controls were analyzed, ROC/AUC analysis showed that the optimal cut-off value for BALF BDG was 290 pg/mL, giving a sensitivity of 75%, specificity of 58%, PPV of 64%, and NPV of 70%; the AUC was 0.70 (95% CI 0.48–0.91).

Of the eight control patients who had BALF BDG values ≥128 pg/mL, two had concomitant Gram-negative bacillary pneumonia caused by *Citrobacter* spp. (BALF BDG 460 pg/mL) and *Stenotrophomonas maltophilia* (BALF BDG 290 pg/mL), which could have led to false positive BDG tests. However, two controls had respiratory tract colonization with *Penicillium* and one other had *Pseudomonas* pneumonia, but their BALF BDG values were 78 pg/mL, 94 pg/mL, and ≤45 pg/mL, respectively.

## 4. Discussion

We evaluated the performance of the BDG assay on BALF as a possible aid for the diagnosis of *Pneumocystis* pneumonia by comparing test results from a cohort with well-characterized *Pneumocystis* pneumonia with a closely matched control cohort. We found that the BALF BDG assay was moderately sensitive for the detection of infection with *Pneumocystis*, but specificity was poor. Our results are consistent with several prior reports with smaller numbers of patients that found BALF BDG to have moderate to high sensitivity, but low specificity for the diagnosis of *Pneumocystis* pneumonia [11,12,13]. In contrast, another study noted both high sensitivity and high specificity when BDG was tested using BALF that was obtained from patients with HIV, who presumably might have had a larger burden of organisms [2].

We had postulated that quantifying BALF BDG when values were found to be ≥500 pg/mL would show clear differences between patients with *Pneumocystis* pneumonia, especially those with proven and probable infection, and control patients. However, we did not find that to be the case, contrasting with results from another study in which BALF BDG levels were significantly higher in patients with *Pneumocystis* pneumonia when compared with colonized and uninfected patients [13].

Specificity is a known drawback with BDG testing, as noted in two meta-analyses dealing with serum BDG assays for *Pneumocystis* pneumonia [5]. This appears to be an even greater problem when BALF is tested, as noted in our study, and reported by others [12]. The reasons for false positive results are related, in part, to the sharing of similar glucans among various fungi and to similar substances on cellulose membranes, gauze packing material, and Gram-negative bacterial cell walls [14]. Why specificity is lower in BALF than serum is not clear. Although two control patients with high BDG levels had evidence of Gram-negative bacterial pneumonia, three other controls with either Gram-negative bacterial pneumonia or fungal organisms colonizing their tracheobronchial tree did not have BDG levels above the cutoff of 128 pg/mL.

In spite of the fact that BALF BDG has not been approved as a diagnostic tool for *Pneumocystis* pneumonia, this test is frequently ordered, many times leading to confusion and erroneous interpretation. We undertook this study, in part, because of this observation. We sought to objectively evaluate the role of BALF BDG for the diagnosis of *Pneumocystis* pneumonia by comparing results obtained from cases that were strictly characterized as proven, probable, or possible *Pneumocystis* infection, with results from tightly matched control subjects.

Limitations of our study include its single-center, retrospective design and the relatively small number of *Pneumocystis* pneumonia cases studied. However, with the exception of a study among patients with HIV [6], other studies of BDG measurements in BALF have each reported on fewer than a dozen patients with *Pneumocystis* pneumonia. Given the retrospective nature of the study, we were not able to control when the BALF sample was taken in the course of the patient’s disease. It is possible that the BDG concentration could rapidly decrease, leading to low levels at the time of sampling; the kinetics of BDG in BALF in the course of *Pneumocystis* infection are not known. Lastly, we cannot predict whether the freezing and then thawing of the BALF specimens had an effect on the performance of the BDG assay.

In conclusion, we have shown that the BDG assay in BALF had moderate sensitivity for *Pneumocystis* infection, but the unacceptably high false positive rate precludes its use as a diagnostic tool for *Pneumocystis* pneumonia.

## Figures and Tables

**Figure 1 jof-06-00200-f001:**
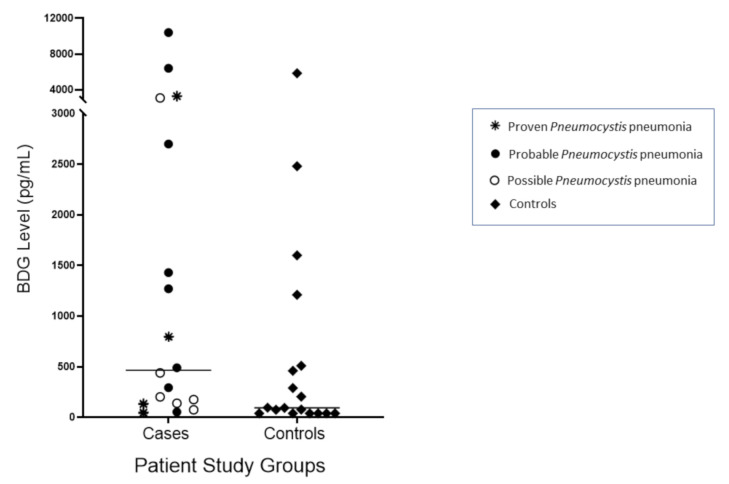
Bronchoalveolar lavage fluid (BALF) (1,3)-β-d-glucan (BDG) concentrations expressed as pg/mL in patients with *Pneumocystis* pneumonia characterized as proven, probable, and possible *Pneumocystis* infection (cases) and matched control patients. Bars represent the median BDG level for that cohort.

**Figure 2 jof-06-00200-f002:**
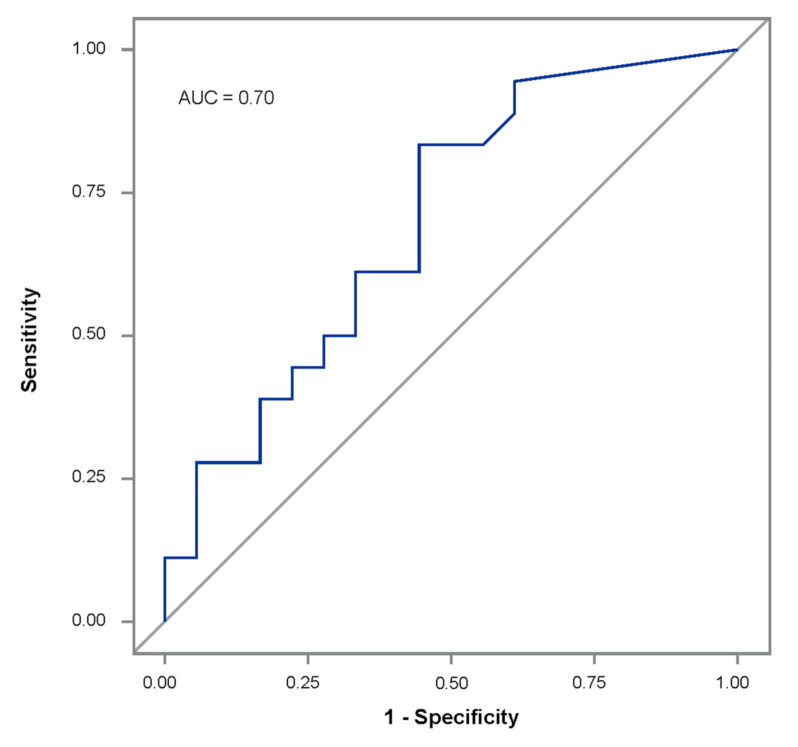
Receiver operating characteristic/area under the curve (ROC/AUC) analysis of bronchoalveolar lavage fluid (1,3)-β-d-glucan (BDG) concentrations to distinguish *Pneumocystis* pneumonia cases from matched controls, AUC 0.70 (95% C.I. 0.52–0.87).

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
