# Peer review of "Diagnostic Performance of Bronchoalveolar Lavage (1,3)-β-d-Glucan Assay for Pneumocystis jirovecii Pneumonia"

_jof, 2020, doi:10.3390/jof6040200_

Round 1
Reviewer 1 Report
Dear authors,
This study is interesting and well written.
Nevertheless, I advise you to correct or add some details:
•Is it possible to add some details about Patients with positive PCR but excluded (number, medical background especially, serum b D glucan values)?
•line 82-86, please provide more details about the PCR and qPCR assay (extraction process of BAL, target???)
•line 99-100: please add details about the dilution approach used for samples > 500 pg/mL
•180: please add as a possible limitation , that you can't exclude an impact of freezing/thawing cycle of BALFs on B D glucan assay
For samples with qPCR results ( after february 2017) , did you observe a correlation between Cq values and BALFs B-D-Glucan values?
Did patients beneficiate from a serum B D Glucan assay ? if yes, do you have some observation about an eventual correlation between serum and BALfs B D glucan values for the Study versus control group?
Sincerely,
Author Response
Comment 1: “Is it possible to add some details about Patients with positive PCR but excluded (number, medical background especially, serum b D glucan values)?”
Answer to comment 1: unfortunately, those patients were excluded and we do not have further data on them.
Comment 2: “please provide more details about the PCR and qPCR assay (extraction process of BAL, target???)”
Answer to comment 2:
Manuscript was revised accordingly and this information was added in lines 114-127 “Endpoint PCR: BALF specimens collected prior to February 1, 2017, were tested by an inhouse Pneumocystis endpoint PCR assay. The target amplified in this procedure was the mitochondrial ribosomal RNA gene rnl (large subunit ribosomal RNA). The amplification product was visualized after being run on an agarose gel and then stained with ethidium bromide. To be called positive, a band had to be present at 346 base pairs. Positive and negative controls were always concomitantly tested with patient samples to ensure the test’s validity.
Real-Time PCR: Beginning Feb 1, 2017, the EliTech InGenius PCR assay (EliTech Group, Bothell, WA) was used for the detection of P. jirovecii DNA in BALF samples. The target amplified in this procedure was the mitochondrial ribosomal RNA gene MtLSU. Specimen extraction, purification of nucleic acids, amplification, and detection of Pneumocystis DNA were all performed by the InGenius instrument. Readouts provided by the instrument showed amplification curves and cycle threshold values. Positive and negative.”
Comment 3: “please add details about the dilution approach used for samples > 500 pg/mL”
Answer to comment 3: Manuscript was revised accordingly and this information was added in lines 115-116. “if reading was > 500 pg/mL further serial 1:10 dilutions were performed in reagent grade water.”
Comment 4: “please add as a possible limitation, that you can't exclude an impact of freezing/thawing cycle of BALFs on B D glucan assay.”
Answer to comment 4: Manuscript was revised accordingly and this information was added in lines 200-201. “Lastly, we cannot predict whether the freezing and then thawing of the BALF specimens had an effect on the performance of the BDG assay.”
Comment 5: “For samples with qPCR results (after february 2017), did you observe a correlation between Cq values and BALFs B-D-Glucan values?”
Answer to comment 5: Unfortunately, that information was not provided to us by the laboratory. At our institution, PJP PCR results are reported as “positive” or “negative”
Comment 6: “Did patients beneficiate from a serum B D Glucan assay? if yes, do you have some observation about an eventual correlation between serum and BALfs B D glucan values for the Study versus control group?”
Answer to comment 6: Unfortunately, serum BDG was performed in only few patients and therefore, this can’t be analyzed.
Reviewer 2 Report
The manuscript presents a retrospective study concerning the usefulness of quantitative evaluation of BDG in BALF for diagnosis of Pneumocystis jirovecii pneumonia. The major sections of the paper are well structured and all data are accurately presented.
The "Introduction" section could be improved by including more literature data and references.
Author Response
Comment 1 - The "Introduction" section could be improved by including more literature data and references.
Answer to comment 1: For this short, very directed manuscript, we feel that the introduction is long enough and provides all necessary references.